# Crystallisation Phenomena of In_2_O_3_:H Films

**DOI:** 10.3390/ma12020266

**Published:** 2019-01-15

**Authors:** Ruslan Muydinov, Alexander Steigert, Markus Wollgarten, Paweł Piotr Michałowski, Ulrike Bloeck, Andreas Pflug, Darja Erfurt, Reiner Klenk, Stefan Körner, Iver Lauermann, Bernd Szyszka

**Affiliations:** 1Institute of High-Frequency and Semiconductor System Technologies, Technical University Berlin, Einsteinufer 25, 10587 Berlin, Germany; s.koerner@tu-berlin.de (S.K.); bernd.szyszka@tu-berlin.de (B.S.); 2Institute for Nanospectroscopy, Helmholtz-Zentrum Berlin für Materialien und Energie GmbH, Albert-Einstein-Str. 15, 12489 Berlin, Germany; alexander.steigert@helmholtz-berlin.de; 3Department of Nanoscale Structures and Microscopic Analysis, Helmholtz-Zentrum Berlin für Materialien und Energie GmbH, Hahn-Meitner-Platz 1, 14109 Berlin, Germany; wollgarten@helmholtz-berlin.de (M.W.); bloeck@helmholtz-berlin.de (U.B.); 4Institute of Electronic Materials Technology, Wolczynska Str. 133, 01919 Warsaw, Poland; Pawel.Michalowski@itme.edu.pl; 5Fraunhofer Institute for Surface Engineering and Thin Films IST, Bienroder Weg 54e, 38108 Braunschweig, Germany; Andreas.Pflug@ist.fraunhofer.de; 6PVcomB, Helmholtz-Zentrum Berlin für Materialien und Energie GmbH, Schwarzschildstr. 3, 12489 Berlin, Germany; darja.erfurt@helmholtz-berlin.de (D.E.); klenk@helmholtz-berlin.de (R.K.); iver.lauermann@helmholtz-berlin.de (I.L.)

**Keywords:** In_2_O_3_:H, thin films, crystallisation, TCO, high mobility

## Abstract

The crystallisation of sputter-deposited, amorphous In_2_O_3_:H films was investigated. The influence of deposition and crystallisation parameters onto crystallinity and electron hall mobility was explored. Significant precipitation of metallic indium was discovered in the crystallised films by electron energy loss spectroscopy. Melting of metallic indium at ~160 °C was suggested to promote primary crystallisation of the amorphous In_2_O_3_:H films. The presence of hydroxyl was ascribed to be responsible for the recrystallization and grain growth accompanying the inter-grain In-O-In bounding. Metallic indium was suggested to provide an excess of free electrons in as-deposited In_2_O_3_ and In_2_O_3_:H films. According to the ultraviolet photoelectron spectroscopy, the work function of In_2_O_3_:H increased during crystallisation from 4 eV to 4.4 eV, which corresponds to the oxidation process. Furthermore, transparency simultaneously increased in the infraredspectral region. Water was queried to oxidise metallic indium in UHV at higher temperature as compared to oxygen in ambient air. Secondary ion mass-spectroscopy results revealed that the former process takes place mostly within the top ~50 nm. The optical band gap of In_2_O_3_:H increased by about 0.2 eV during annealing, indicating a doping effect. This was considered as a likely intra-grain phenomenon caused by both (In^0^)_O_^••^ and (OH^−^)_O_^•^ point defects. The inconsistencies in understanding of In_2_O_3_:H crystallisation, which existed in the literature so far, were considered and explained by the multiplicity and disequilibrium of the processes running simultaneously.

## 1. Introduction

Hydrogen doped indium oxide (In_2_O_3_:H) films demonstrate an electron Hall mobility (μ*_e_*) of over 100 cm^2^/Vs [1,2]. Such outstanding property can be achieved if the In_2_O_3_ film is deposited in an amorphous state and subsequently crystallised at T > 160 °C. Its optical transmittance is superior in both ultraviolet (UV) and near-infrared (NIR) spectral ranges when compared to the widely used transparent conductive oxides (TCOs), such as ZnO:Al or In_2_O_3_:Sn. This results in a performance gain for Si-based [3,4,5] and CIGS (Cu-In-Ga-S/Se)-based [6,7,8] solar cells.

Amorphous In_2_O_3_ films are usually obtained by sputter-deposition in the presence of water vapour without intentional heating [1,2]. Low crystallisation temperatures make them applicable on top of a-Si and CIGS-absorbers. Moreover, the crystallisation seems to proceed within a few minutes [3]. Thus In_2_O_3_:H can be essential for tandem solar cell concepts, e.g., with hybrid perovskites.

There is a number of thorough theoretical and experimental investigations on In_2_O_3_, its defectiveness and interaction with a gas phase. Nevertheless, some ambiguity with respect to the origin of the *n*-type conductivity, the role of the crystallisation and the origin of the high carrier mobility In_2_O_3_:H films still exists. For instance, there is no satisfactory explanation for the following phenomenon so far: the crystallisation of In_2_O_3_:H films is accompanied with a Burstein-Moss shift and a decrease of the charge carrier concentration (*N*_e_) simultaneously. At the same time, it is widely accepted in literature that the *n*-type doping of crystalline In_2_O_3_ is related to the hydrogen incorporation. Another important disagreement concerns the “high mobility”. Exactly the passivation of dangling bonds on grain-boundaries by hydrogen is supposed to be the most probable reason for high mobility [9]; however, only this intergranular hydrogen disappears primarily during heating in vacuum [10].

Our recent studies revealed the effect of oxygen addition to the sputter-gas on the In_2_O_3_:H crystallisation progress [11]. However, the question of whether we deal with so-called plasma damage still remains. This negative effect of the accelerated O^−^ ions largely determines electrical properties and their spatial deviations over the film [12,13].

The inconsistencies noted above represent a lack of understanding of the In_2_O_3_:H material. In this work we investigated microstructural, compositional, optical and electrical properties of the In_2_O_3_ and In_2_O_3_:H films. We also observed how these properties change during In_2_O_3_ crystallisation. Additionally, the discussion section contains a brief review of the relevant findings accumulated so far in literature. Finally, we attempted to compose a full puzzle from all these data and understand this material better.

## 2. Experimental Section

The as-deposited films will be further assigned as In_2_O_3_:H_2_O, because they were found to contain hydroxyl groups. The post-deposition annealed films we denote by In_2_O_3_:H, as widely accepted. In_2_O_3_:H_2_O films were obtained in a stationary mode by RF (Radio Frequency, 13.56 MHz) magnetron sputtering from a round Ø 3″ ceramic In_2_O_3_ (99.99%) target onto unheated, stationary 1″ × 1″ Eagle XG (Corning, 0.7 mm thick) glass substrates. The substrates were cleaned in a multistage glass-washer using surfactants and de-ionised water. The target-to-substrate distance was fixed at the optimum 40 mm. The RF power was set at 60 W, which results in 1.3 W/cm^2^ power density. The base pressure of the sputtering system was 5 × 10^−6^ Pa. Water vapour was admitted through a needle valve from a reservoir. Adjusting the valve manually, the *p*(H_2_O) was stabilised at 2 × 10^−3^ Pa prior to the deposition. The sputtering gas was pure Ar, and the following process pressures were compared: 0.5 Pa (Ar flow: 20 sccm) and 1.3 Pa (Ar flow: 100 sccm). The films were annealed after deposition under different *p*(O_2_) conditions: ~21 kPa (ambient air) or ~10^−6^ Pa (ultra-high vacuum). Very thin (~20 nm) films obtained on 0.3 mm thick Si-wafers were analysed by XPS/UPS without breaking vacuum in the combined deposition and surface analysis system (CISSY) [14]. It includes a standard XPS laboratory module with a non-monochromatic X-ray source and Mg (Al) anodes providing an excitation energy of 1253.6 eV (1486.6 eV). UPS measurements were done with a standard He lamp, yielding 21.22 eV (He I) excitation energy. As photoelectron analyser, a VG CLAM 4 with a hemispherical energy filter and an electron detector based on discrete channeltrons was operated at 20 eV pass energy. A sputtered Au foil served as a reference for energy calibration based on the Au4f_7/2_ transition at 84.0 eV.

Si-substrates were also used for the secondary ion mass-spectroscopy (SIMS) investigation so to provide better analytical response. The RF plasma effectively heats thin substrates, resulting in uncontrolled crystallisation during deposition (see Appendix A). Therefore we have used DC sputtering to secure the amorphous state of In_2_O_3_:H_2_O film, which should be thick enough for the meaningful depth resolved SIMS. Our long-time experience revealed no principle difference in optical and electrical properties of the RF- and DC-sputtered In_2_O_3_:H films crystallised from a fully amorphous state. A mid-frequency pulsed (65 kHz, τ = 3.2 μs) DC magnetron sputtering was conducted in the deposition system A600V7 (Leybold Optics Dresden GmbH) at 75 mm target-to-substrate distance. The power density was 2.67 W/cm^2^, total pressure—0.4 Pa, partial pressures of oxygen (O_2_) and water (H_2_O) during processing were 2.3 × 10^−4^ Pa and 1.7 × 10^−4^ Pa, respectively. The deposition took place on an oscillating substrate. In_2_O_3_ films were deposited under the same conditions but without introducing water vapour. The annealing took place in UHV at 220 °C for 30 min.

All SIMS depth profiles in this work were performed employing the CAMECA SC Ultra instrument operating under ultra-high vacuum (UHV) of ~4 × 10^−8^ Pa. A Cs^+^ primary beam with an impact energy of 3 keV and intensity of 10 nA was scanned over the (150 × 150) μm^2^ area, whereas the analysis was limited to the (50 × 50) μm^2^ area. The positive ion detection mode was used and thus each element was measured as the CsX^+^ cluster. Subsequently, a point-to-point normalisation to the Cs^+^ signal was performed. Thus, due to a significant reduction of the matrix effect [15,16,17,18,19], we were able to detect adequately the In/O and H/O ratios as well as the annealing losses. The latter were determined as follows:(1)$$L_{H}=1-\frac{I_{H\;annealed}}{I_{H\;as-deposited}},\;L_{O}=1-\frac{I_{O\;annealed}}{I_{O\;as-deposited}}$$
where *I*—are the intensities normalised to the Cs signal and *L***_H_** and *L***_O_** are the relative annealing losses of hydrogen and oxygen, respectively. Despite the reduced detection limit for hydrogen in the positive ion detection mode, its concentration in water containing samples was found to be high enough for a reliable determination.

Fourier-transform infrared spectroscopy (FTIR) was performed on a Vertex 70 (Bruker Optics, Ettlingen, Germany) using the rock solid interferometer, mercury (Hg) IR-source, DLaTGS IR-Detector and the KBr Beam-splitter. In_2_O_3_:H films were deposited on double-side polished Si-substrates for this investigation. XRD patterns were recorded using Cu K_α_ radiation in different scanning modes: symmetrical Bragg-Brentano and asymmetrical detector scan. In the *in-plane* measurement, both incident and diffracted beams had grazing angles to the sample surface. D8 Discover (Bruker, Karlsruhe, Germany) and X’Pert MRD Pro (PANalytical, Almelo, The Netherlands) diffractometers were used for these tasks. LaB_6_ powder (660c) was used as a standard for estimation of the crystallite size. The Hall mobility of charge carriers was measured with an Ecopia HMS-3000 system (Anyang-city, Gyeonggi-do, Korea) in van der Pauw geometry at room temperature. The proportionality factor was taken equal to 1. Scanning electron microscopy (SEM) was made using a Hitachi S4100 microscope (Hitachi High-Technologies Corporation, Tokyo, Japan).

Samples for cross sectional transmission electron microscopy (TEM) were prepared by gluing the thin films face-to-face, followed by mechanical and ion thinning for electron transparency. TEM investigations were performed on two microscopes. The Philips CM12 (FEI Company, Hillsboro, OR, USA) was used at 120 kV accelerating voltage for preliminary investigation. The system Zeiss LIBRA 200 FE (Carl Zeiss Microscopy, Jena, Germany) operated at 200 kV accelerating voltage was used for more detailed analysis by electron energy loss spectroscopy (EELS). This microscope is equipped with an in-column energy filter for energy filtered image acquisition. The set electron energy loss was varied from 0 to 30 eV.

## 3. Results

### 3.1. Crystallinity versus Electron Mobility

The crystallinity of TCO films, i.e., the size of grains and the texture, can determine electron mobility to a different extent, which depends on the individual material. According to the band structure calculations accepted for In_2_O_3_, conduction band minima are formed by the 5 s states [20,21,22]. Because of their spherical symmetry, one may then expect no significant impact of the coherence between grains (texture) onto the grain boundary scattering. However, a limiting role of the grain boundaries themselves (grain size) illustrates the following fact: the undoped single crystalline In_2_O_3_ films and In_2_O_3_ single crystals demonstrate an electron mobility of more than 100 cm^2^/Vs [23,24] whereas the as-prepared polycrystalline films do not reveal μ_e_ exceeding 50–60 cm^2^/Vs [25,26]. This fact is basically related to the difference in *N*_e_, which is known to be much lower in single crystals (~10^17^ cm^−3^) as compared to the polycrystalline materials (10^19^ cm^−3^) [23]. In turn, high *N*_e_ values in polycrystalline In_2_O_3_ are usually attributed to the so called unintentional doping caused by the presence of water in ambient air. Additional negative impact on mobility provide ionized impurities.

In this work, we intended to examine to what extent the grain size and texture determine Hall mobilities in In_2_O_3_:H films. Figure 1 presents the X-ray patterns and Hall mobility data for various as-deposited and annealed films. Here, two series of experiments were done: variation of the film-thickness (*a*) and variation of the total pressure during deposition (*b*). As one may see from the series (*a*), all ≥200 nm thick In_2_O_3_:H_2_O films appear to be X-ray crystalline. Diffraction maxima are, however, quite broad. As we explained above, this is due to the heating of the surface by the RF plasma. This effect increases with time due to the NIR absorption of the growing film and at some point the film becomes crystalline (see Appendix A).

Post-deposition annealing at 180 °C for 60 min in ambient air results in further crystallisation and increase of electron mobility. The biggest gain of mobility is demonstrated by thinner, initially X-ray amorphous, films. All partially crystalline In_2_O_3_:H_2_O films thicker than 300 nm reveal almost the same gain of mobility after annealing. Analysing XRD patterns in Figure 1a, one can find two In_2_O_3_ phases: initially crystalline and crystallised after annealing. A coexistence of two phases is especially well visible in the 400 nm thick film. Correlating this with Hall mobility data, we conclude that exactly the post-deposition crystallisation is crucial for high mobility. Interestingly, the “high-mobility” phase reveals a smaller lattice constant than the phase, which crystallises during deposition. The origin of this difference will be discussed below.

A decrease of intensity of diffraction peaks is observed for the initially crystalline In_2_O_3_ phase as a result of annealing. The most probable reason for that is active recrystallization.

Figure 1b demonstrates the influence of the sputtering pressure on the crystallisation behaviour of X-ray amorphous films. Apparently, this parameter determines the film texture and Hall mobility. We attribute this effect to the amount of hydroxyl in a film. Indeed, using a fixed leak rate of the needle valve feeding water vapour, a higher *p*_tot_ should result in a smaller *p*(H_2_O) and, hence, in a smaller concentration of hydroxyl in In_2_O_3_. In turn, the impact of hydroxyl on the crystalline growth of In_2_O_3_ is a known phenomenon, thus, the (400)-orientation can be suppressed if water is present in a sputtering gas [27]. This is consistent with our results. A higher process pressure itself may also contribute, as it increases the energy transfer from plasma to substrate, resulting in a higher deposition temperature [28].

Considering the cubic bixbyite structure of In_2_O_3_, one can distinguish two kinds of InO_6_ octahedrons which interconnect either by corners or by edges [29]. As the octahedrons are directed along the [111] axes in the lattice, the close packed oxygen layers coincide with the {00*l*} planes. This explains why the <00*l*> oriented films might reveal a smaller electron mobility.

In our further XRD, TEM and SIMS investigations we used the ~150 nm thick films. XRD measurements (see Appendix A) were performed to estimate the size of coherent scattering in both lateral (*D*_lat_) and longitudinal (*D*_long_) directions using Scherrer’s method [30]. In case of lateral size the asymmetric *in-plane* XRD measurements at *ψ* = 89.2° were performed. Assuming single strength and linear *d_hkl_* − sin^2^*ψ* dependence we estimated the residual stress in the films. A complex recalculation which takes into account the measurement conditions was performed on the basis of a procedure explained elsewhere [31]. The elastic constants for In_2_O_3_ were taken from literature [32,33]. Our results are collected in Appendix A and Table 1. Hall measurement data are also presented for the assessment.

We see that the crystalline state itself does not secure high electron mobility: compare, for instance, the crystalline state III with the amorphous state II—both films were deposited in the presence of water vapour. The positive impact of water on μ_e_ is, however, obvious (compare state I with other states). This effect is explained in literature by a passivation of dangling bonds with hydrogen that decreases the electron scattering [9]. One can see that such passivation is highly pronounced exactly in the amorphous state with the utmost amount of dangling bonds (compare state II with I or III). We observe that μ_e_ strongly increases with the grain size, compare states III → (IV, V). Thus, both qualities: the degree of crystallinity and the passivation of dangling bonds are important for reaching a high electron mobility in In_2_O_3_.

It should be noted why the concentration of free electrons changes. The presence of water does not result in a marked change of *N*_e_ (compare states I and II), but high temperature does. This effect will be discussed below.

One has to notice the presence of residual compressive stress in crystalline films, especially if crystallisation takes place during deposition (states I and III). The values presented are just an estimation with a large error, based on the measurements of only two *ψ* values. Two reflexions (222) and (400) with quite similar Poisson’s ratio were taken into account [33].

It is a fact that in the presence of water the growing In_2_O_3_ films contain hydroxyl groups [9]. Moreover, the hydroxylation apparently stipulates the amorphous state of In_2_O_3_ [34]. We have, however, made some curious observations in our experiments, which cannot be easily explained. Thus, the In_2_O_3_:H_2_O films (state II in the Table 1) grow predominantly X-ray crystalline on such substrates like molybdenum-films, *i*-ZnO, Si-wafer, or CIGS-absorber [11]. Moreover, if any additional oxygen is injected into the sputtering gas, the films becomew crystalline and resistive.

### 3.2. Presence and Role of Metallic Indium

Since the films appearing as X-ray amorphous could still be nano-crystalline, we undertook TEM investigations of them. Figure 2 shows cross-sectional brightfield TEM images of an In_2_O_3_:H_2_O film deposited at *p*_tot_ = 0.5 Pa on glass.

Several effects can be observed. Initially, the film is amorphous, but it changes under the electron-beam after about 20 min. A diffraction contrast becomes visible, indicating crystalline structures. At the same time, droplet-like features at the film/glass interface increase in number and size. Being aware of the inherent effects of our fabrication procedure, we presume here an effect of the sample heating by the e-beam. Basically, heating of TEM-lamella due to inelastic interactions with high energy electrons is a known phenomenon [35]. According to the literature, relatively thick In_2_O_3_:H films crystallise at 180–200 °C [1,2,3]. In our special case we deal with much thinner lamella, where the impact of surface and contact interface is apparently larger. This fact may shift the crystallisation temperature to lower values. Apart from the crystallisation effect observed, we assume that the appearance of droplet-like features is related to the presence of metallic indium. Its melting (*T*_m_ = 156.6 °C) might promote In_2_O_3_:H_2_O crystallisation and results in accumulation of In-droplets.

Actually, the appearance of metallic indium seems to be rather probable in our case as the metallic phase was found in In_2_O_3_ and In_2_O_3_:Sn films by other investigators [36,37]. Consideration of the In-O phase diagram and general chemistry of indium oxide/hydroxide system also supports this assumption [34,38].

We applied energy filtered TEM to detect metallic indium, as it exhibits a bulk plasmon with an energy of about 12 eV [39]. The plasmonic spectrum of In_2_O_3_ [40] could not be observed in this study. The set of TEM images obtained at distinct loss energies is presented in the Appendix A. Metallic indium appears as bright areas at an electron energy loss of ~12 eV. We also found that indium congregates either at the film-glass interface or localises in the bulk (see Appendix A). The latter case is shown in more detail in Figure 3, where the correlation of crystallinity (Figure 3a) and appearance of metallic indium (Figure 3b) can be observed. The top In_2_O_3_ layer grows crystalline due to the impact of RF plasma as we discussed above. The crystalline part consists of columnar crystallites of 20–40 nm lateral size, which is consistent with the film state III in Table 1. One can conceivably detect some porosity within this area (see Figure 3a and Appendix A). Metallic indium particles segregate exactly at the transition region between amorphous and crystalline layers (Figure 3b). These nanoparticles were found to be crystalline (Appendix A). Obviously, melting of indium and hence its extraction in a separate phase is associated with indium oxide crystallisation. The liquid phase is known to support even high quality crystallisation in such methods as liquid phase epitaxy [41], metal-modulated epitaxy [42], volatile surfactant assisted chemical vapour deposition [43] and others.

### 3.3. Optical Properties of In_2_O_3_:H_2_O and In_2_O_3_:H

Optical measurements allow determining properties of the continuous matter. If a material is crystalline, we receive the information from the interior area of grains, whereas the electrical properties are cumulative. Many optical investigations of In_2_O_3_ and In_2_O_3_:H films have been undertaken [2,36,44,45]. According to S. Joseph and S. Berger, the fitting of IR optical transmission spectra by effective medium approximation reveals that no changes in In_2_O_3_ transmittance should be observed if the volume fraction of metallic indium remains less than 10%. A larger indium excess results in a markedly lower transmittance as compared to that experimentally observed [36]. The presence of a metallic phase can be also deduced from the temperature dependence of the resistivity [46]. This, however, demands an even higher volumetric content of indium for the percolation of electrical current.

Figure 4 presents fitted optical spectra for the thin films. The fits were obtained with the help of the RIG-VM software developed at Fraunhofer IST [47]. A Tauc-Lorentz oscillator has been used for the fundamental absorption and a Drude term for the free electrons. A free electron mass of *m** = 0.28 *m_e_* has been used [23]. As one can see, the optical mobility matches well with the Hall mobility (compare with Table 1) for both film states. Some deviation is observed for the crystalline state, where the Drude term gives a somewhat smaller mobility. This can be attributed to the insufficient spectral range to fit the plasma resonance of free carriers accurately. A remarkable difference in *N*_e_ values should be noted, namely, the electrical measurements gave an approximately double concentration of free electrons, which are not visible optically. This means that we might observe an additional inter-grain source of free electrons.

Based on the measurements in the UV range, we determined optical band gaps based on the Tauc-Lorentz model. These values are presented in Table 2 for the material states I, II and V.

According to the literature, there cannot be an indirect gap in pure In_2_O_3_ due to the parabolic nature of the conduction band [22]. Moreover, the minimum band gap at 2.9 eV is symmetrically forbidden and the first allowed optical transition occurs from the level ~0.8 eV below the valence band top that gives the commonly observed value of *E*_g_ ≈ 3.7 eV. As presented in Table 2, the as-deposited crystalline In_2_O_3_ films (state I) reveal considerably lower *E*_g_ as compared to the expected value for this material. Amorphous films, which probably contain water (state II), demonstrate even lower values; however, for 400 nm thick films the difference between states I and II is negligible. This is consistent with the fact of partial crystallisation observed in thicker In_2_O_3_:H_2_O films (see Figure 1 and Figure 3). However, no influence of water in the state II is then noticed. We do not detect the influence of expected hydrogen doping in the post-crystallised In_2_O_3_:H films (state V) as well, because the optical *E*_g_ values observed are very close to the ones known for the pure In_2_O_3_.

### 3.4. Chemical Changes in In_2_O_3_

To understand if there is any chemical transformation during crystallisation and which doping mechanism is realised, we performed further investigations. According to the FTIR analysis (data are not presented in this paper) none of the OH or adsorbed water (1615–1630 cm^−1^) features were observed in crystalline films. This might mean that these species, if they exist, are concentrated mainly at the grain boundaries. The IR transmittance, however, differs considerably for different film states. Thus, an addition of water during deposition results in a significant reduction of the IR transmittance (states I and II are compared). Annealing in vacuum does not significantly change it (states II → V), whereas annealing in ambient air provides quite a strong increase of the IR transmittance (states II → IV). Taking into account our TEM/EELS results, we attribute this behaviour to the presence of metallic indium in the films deposited in the presence of water. Annealing of such films in ambient air provides more effective oxidation of indium as compared to the annealing in vacuum.

SIMS was used to determine the In/O and H/O ratios across the film. As we pointed out in experimental section the DC-sputtered~150 nm thick In_2_O_3_ films were analysed in this case. Figure 5a compares the In/O ratio in as-prepared and annealed films deposited with and without water. As one can see, In_2_O_3_ films contain less oxygen and do not change during annealing in vacuum. According to our visual observations, In_2_O_3_ films are usually darker than In_2_O_3_:H films. Therefore, we assume the In_2_O_3_ to be oxygen deficient. On the contrary, very transparent In_2_O_3_:H films seem to possess a stoichiometry close to *x* = 3 in In_2_O*_x_*. Curiously, we observe the same oxygen enrichment profile in the upper ~50 nm of both: as-deposited and annealed In_2_O_3_ films. At the very surface this enrichment approaches the overall level observed in the annealed In_2_O_3_:H film. As this effect is insensitive to annealing, we most probably deal with the impact of a short exposure to the ambient air prior to the measurement. It is observed only in the case of tiny crystalline, sub-stoichiometric films, which indicates a reaction in the inter grain space.

Water containing, as-deposited films (state II) have the highest oxygen content among the investigated systems. It decreases after annealing (state V) together with the increase of transparency in both UV and near IR spectral regions. We attribute this oxygen loss to the release of water, as the In/O ratio remains unchanged in the water-free In_2_O_3_ sample under the same treatment. Furthermore, active oxygen diffusion in In_2_O_3_ starts at temperatures above 600 °C [48].

For better understanding, we represented the measured data in the form of an H/O ratio (Figure 5b) and the percentage loss of hydrogen and oxygen (see formulas (1)) as a result of the annealing (Figure 5c). Obviously, hydrogen-to-oxygen ratio is higher in as-deposited state. Its depth profile demonstrates several pronounced maxima, which correspond to the substrate oscillation and passing by the inlet of water vapour. This result validates our procedure of hydrogen detection, proving the satisfactory sensitivity, which we can only achieve for the water-containing films. On the other hand we realise that hydrogen detected in as-deposited films represents most likely just water. A comparison of *L*_H_ and *L*_O_ discloses an interesting effect: the oxygen loss remains stable over the entire film thickness, whereas hydrogen releases more actively from the top but demonstrates a stable *L*_H_ in the depth. We suggest the release of H_2_O and H_2_ species from the film being annealed (Figure 5c), as only their formation in a free volatile form is chemically possible. Water can evaporate in its free form if it is contained or released from the hydroxyl groups as shown in the reaction (4) below. Hydrogen would form only in the presence of metallic indium according to the reactions:In + 3 H_2_O → In(OH)_3_ + 3/2 H_2_(2)
In + In(OH)_3_ → In_2_O_3_ + 3/2 H_2_(3)

The probability of such reactions and some supporting experimental data published will be considered in the discussion chapter below.

Considering SIMS results, we could not operate with the absolute values since we did not use any external standard. However, the qualitative suggestions made were based on internal standards—indium and oxygen. We realised that the crystallised In_2_O_3_:H film, which is a high mobility TCO to be used in various devices, might suffer from the chemical heterogeneity.

To observe the processes taking place on the film surface, UPS and XPS measurements were undertaken. Our XPS measurements (spectra are not shown here) reproduced the results obtained by Hans F. Wardenga, where In_2_O_3_:H_2_O films revealed a shoulder at the O1s emission line at about 532.6 eV binding energy [9]. This was found to correspond to the OH bonds, which disappear after annealing.

The UPS spectra acquired during a stepwise increase of temperature from the ambient level (~25 °C) up to 230 °C in UHV show the following changes in the In_2_O_3_:H_2_O film (Figure 6). A ~0.4 eV shift in the secondary electron edge is observed. The secondary electron edge can be used to determine the work function of a material according to the relation *W_f_* = *E*_ex_ − *EB*_sec_. Thus, we observe here the *W_f_* change from ~4.0 eV to ~4.4 eV that basically contradicts the doping phenomenon. It is worth noticing that the work function of the thermally deposited fully oxidised indium oxide is 5.0 eV [49]. This means that indium in the films in question has a lower oxidation state than in the stoichiometric oxide. The value of *W_f_* is also determined to a large extent by the crystallographic orientation [50]. In our case the observed increase of a work function is probably caused by the appearance of a distant order. The significant difference relative to the fully oxidised In_2_O_3_ state can be connected to the presence of metallic indium in the films (our TEM and FTIR data).

According to the Figure 6b, vacuum annealing causes a shift of the valence band edge by ~0.2 eV. All observed changes are be depicted on the energy diagram, where the UPS data are used to fix the *E*_F_ and *E*_VB_ levels (Figure 7). Here we used a caption *E*_g, min_ (minimum) for the fundamental band gap, which is known to be 2.9 eV for the pure In_2_O_3_ [23]. If we admit any doping in our films, it can be even smaller due to the band gap narrowing phenomenon [51,52]. The optical *E*_g, opt_ values obtained in this study were placed in accordance with the principle described above [22]. These energy diagrams show that the Fermi level is very close to the conduction band in both materials: amorphous and crystalline. If we admit the same fundamental band gap width for both states, then the latter would be a non-degenerate semiconductor that should not be the case for such high concentration of free electrons. Since we observed a Burstein-Moss shift as a result of the annealing, the In_2_O_3_:H likely remains degenerate due to the band gap shrinkage. Major changes happen with a level of the allowed optical transition inside the valence band, namely, it shifts markedly downwards. This can be attributed to the effect of crystallisation as the valence band contains fully occupied 2*p* and 2*s* oxygen states and empty 4*d*-indium states [22].

The observations made require a more detailed discussion of the In_2_O_3_ chemistry and possible origin of doping.

## 4. Discussion

To understand the results obtained in this work, we have to review some basic properties of indium, In_2_O_3_ and In(OH)_3_.

### 4.1. Appearance of Metallic Indium in In_2_O_3_

The electro-chemical potential of metallic indium is *φ*^0^ = −0.3382 V [53]. This means that under normal conditions, metallic indium should not reduce protons in an aqueous solution to molecular hydrogen. Nevertheless, the potential is not too high and both reactions, reduction of metal and reduction of hydrogen, may proceed simultaneously on a competitive basis.

According to the In-O phase diagram, there is no detectable phase of oxygen non-stoichiometry [38]. If any In-excess is provided (≥0.02 at.%), there is a mixture of two phases: In_2_O_3_ and metallic In, which is solid below 156.634 °C and liquid above this temperature. Indium (III) oxide is thermodynamically very stable over a wide range of *T* and *p*(O_2_). According to the Ellingham diagram, one needs a *p*(O_2_) of about 10^−100^ atm. in order to reduce it to metallic indium at room temperature. The equilibrium oxygen partial pressure at 200 °C is about 10^−55^ atm. At the same time, metallic indium remains stable in air and starts to oxidise visibly only after melting. The oxidation of indium in a liquid form proceeds about five time faster as compared to the solid [54]. Aside from the main oxidation state +3, indium may have also +2 and +1 in combination with oxygen. A formal oxidation state +2 is most probably a mixture of diamagnetic +1 (5*s*^2^) and +3 (5*s*^0^) forms, as no experimental evidence of magnetism in reduced indium oxides was detected. The theoretical investigation of hypothetical neutral, molecular In-O clusters with different In/O ratios reveals their high instability in an ionic environment [55]. The HOMO–LUMO gap was found to depend on the metal-to-oxygen ratio in the cluster. Oxidation is likely unfavourable when the In/O ratio is larger than 1, as both vertical and adiabatic electron affinities are negative for In_2_O. In oxygen-deficient In_2_O_3_ films, metallic indium may form as a result of oversaturation by cooling down after deposition at elevated temperature [36]. In this case, indium precipitates according to HRTEM and EELS in a form of 5–30 nm nano-particles independently of an indium excess. It is known that intermediate oxides disproportionate in contact with water, resulting in In_2_O_3_ and metallic indium [34]. These data confirm that metallic indium readily forms if any lack of oxygen and/or water is provided.

To understand the chemical impact of water during sputtering, let us briefly consider the plasma chemistry of water. Basically, low total pressure and especially plasma excitation change the chemical activity of water. Upon photo-ionisation, water vapour becomes a weakly ionised plasma consisting of electrons and H_2_O^+^ [56]. In the highest state of excitation, the plasma consists of *e*^−^, H^+^, and O(*^n^*^+^). In a general case of RF-sputtering from the ceramic target, mostly M^+^ and MO^+^ charged species are observed [57]. In the case of In_2_O_3_, the RF plasma should contain these species in a ratio M^+^/MO^+^ of more than 30. Since this ratio depends on the M-O binding energy, we took the value known for Fe_2_O_3_ implying that the M-O binding energy is quite close for Fe_2_O_3_ (Δ*G*°_298_ = −732.1 kJ/mol) and In_2_O_3_ (Δ*G*°_298_ = −809.3 kJ/mol) [58,59]. When argon is used as a sputtering gas, almost no O^+^, but mostly neutral oxygen is observed [58,60]. The RF-plasma above the ZnO target has a similar content; however, oxygen species generated during DC sputtering of ZnO are O^−^, O_2_^−^, and O. The content of negatively charged oxygen species increases exponentially with the reduction of the total pressure [61]. The main difference between the RF and the DC process lies in the concentration of electrons, which is much higher in the first case. Thus, in our process, we likely deal with an intermediate oxidation state of indium in the absence of strong oxidants in the plasma, which finally yields metallic indium species in a film.

### 4.2. Water Containing In_2_O_3_

Despite a chemical impact as hydroxylation, water stipulates the amorphous state of as-deposited films. It is known that indium (III) hydroxide tends to remain jelly or even forms a colloid in aqueous solutions rather than precipitating in a crystalline form. The main reasons for that are the donor-acceptor interaction, typical for metals having free 3*d* orbitals, and the hydrogen bonding in hydroxides. As it is known for the most investigated analogue—aluminium hydroxide, such parameters as concentration, temperature and pH determine the hydrolysis, peptisation, aging and, finally, crystallisation [62]. Basically, al least three processes are coupled with water release and formation of many networking chemical bonds:=In–OH + HO–In= → =In–O–In= + H_2_O(4)

According to the thermal gravimetric analysis, the crystalline In(OH)_3_ transforms into In_2_O_3_ with water elimination, starting very slowly from T ≥ 200 °C and becomes fast at about 230 °C in an inert 1 bar atmosphere [63]. As per Le Chatelier’s principle, the dissociation in vacuum likely proceeds at lower temperature.

It is known that gaseous hydrogen can also be successfully applied as a hydrogenation agent yielding high-mobility In_2_O_3_ films [64]. In this case, the films were obtained by RF sputtering in an amorphous state as well and crystallised by post-deposition annealing. Thorough investigation of oxygen and hydrogen desorption from the In_2_O_3_ powders with different surface areas serves us with the following observations [10]. Surface hydrogen starts to desorb in a high vacuum (*p* = 5 × 10^−7^ mbar) already at temperatures somewhat below 100 °C. Desorption of stronger bound hydrogen starts at ~150 °C. Water (6 mbar in 1 bar He) becomes an active re-oxidation agent at temperatures higher than about 250 °C, whereas dry oxygen (1 bar) actively re-oxidises the surface starting from >150 °C.

Thus we realise that hydroxylation of In_2_O_3_ is most likely the reason for the amorphous state of as-deposited films. This effect can be achieved via sputtering in the presence of either hydrogen or water. Hydrogen acts even more reproducibly [64], since it probably delivers just a necessary hydroxylation without any water excess. However, we still need to understand the desorption of chemically different hydrogen. Additionally, the effect of In_2_O_3_ reduction in hydrogen on its electrical conductivity should be considered.

### 4.3. Doping and Conductivity of In_2_O_3_

It is known that oxygen deficiency in In_2_O_3_ causes higher conductivity [10,46,65]. According to the impedance measurements performed on In_2_O_3_ polycrystalline samples, their reduction in dry hydrogen results in a slow resistance decline, starting already at room temperature. The resistance falls sharply at a temperature somewhat below 100 °C and further decreases much slower up to its minimum at about 250 °C [10]. In the presence of water, the same dependency is observed, but the temperatures are about 50 °C higher. This change is reversible and matches the hydrogen adsorption/desorption data; however, the resistance of the re-oxidised samples was found to be 4–5 orders of magnitude lower than the initial one. It might point to the inter-grain changes, e.g., In_2_O_3_ reduction, which is then encapsulated by the fully oxidised shell.

We already showed above that the concentration of electrons and hence electron mobility in In_2_O_3_ films were often found to be determined by the size of crystallites [23]. In the literature this effect is attributed to the so-called unintentional doping, which is supposedly caused by the inter-grain diffusion of water from ambient air [66,67]. The mechanism of such doping, however, remains questionable for us.

Theoretical studies of this matter demonstrate quite discrepant conclusions. Some basic description of the defect chemistry in In_2_O_3_ was done, using solid state chemistry [68]. However, most modern investigations being aimed to justify which point defects exist in the material are performed using density functional theory (DFT). Thus, according to J. Liu, who used the GGA + U formalism, the most stable point defects in In_2_O_3_ crystals are oxygen vacancies of the anti-Frenkel type (V_O_^••^-O*_i_*″) [69]. According to the LDA and LDA+U functional calculations, the formation energy of V_In_‴ was found to be very low in *n*-type In_2_O_3_ [70]. Considering hydrogen doped In_2_O_3_, S. Limpijumnong with co-workers suggested H*_i_*^•^ and H_O_^•^ as the main donor defects in In_2_O_3_:H rather than V_O_^••^ [71]. A combination of theory with the muon rotation/relaxation spectroscopy revealed that the charge neutrality level (CNL) for hydrogen in In_2_O_3_ lies above the conductive band minimum (CBM), thus providing a shallow donor level with an activation energy of 47 ± 6 meV [66]. This study also stated that hydrogen often becomes an unintentional donor in many polycrystalline oxides. According to T. Tomita et al., who used first-principles molecular orbital calculations, the interstitial indium ions (In*_i_*^•^) are the native donors in In_2_O_3_ [72]. These defects may only coexist with V_O_ (no charge was noticed in the original work), which facilitate the emergence of indium donors as shallow states.

Considering the penetration of hydrogen into indium oxide, we have to take into account the classical approach of experimentally obtained ionic radii. Oxygen ions (O^2−^) with tetrahedral coordination, like in the In_2_O_3_ bixbyte structure, have an effective ionic radius of 1.38 Å [73]. The OH^−^ group would have an even smaller (1.35 Å) ionic radius in this coordination, since the proton is actually a pristine positive nucleus, which is drawn in to the negative electron shell of oxygen, thus making the Coulomb repulsion between neighbouring oxygen ions smaller. On the basis of this simple consideration it is hard to imagine that the hydrogen proton or even the neutral H atom, having a Bohr radius *a*_0_ ≈ 0.53 Å, can replace oxygen in its site. Thus we recognise that the hydroxyl (OH^−^)_O_^•^ is the most probable hydrogen containing species in In_2_O_3_. It is known, however, that In^+3^ also shows the atomic absorption of hydrogen [74]. Hydrogen was also found to be readily adsorbed by an indium rich InP surface [75]. A large thermodynamic driving force for the neutral covalent binding between hydrogen and solid state indium dimers was identified.

The existence of interstitial indium ions has also some restrictions. Thus, In^+3^ in octahedral coordination possesses, an ionic radius of 0.8 Å is rather large to squeeze into the cavities of the bixbyite lattice. A lower indium oxidation state, e.g., +1 (In*_i_*^•^), corresponds to an even larger ionic radius. The intermediate oxidation states are furthermore electrochemically unstable (see above). However, ferromagnetism was observed in oxygen deficient InO_x_ films annealed in UHV at 600 °C [65]. This effect was found to be accompanied by the In-In clustering and formation of highly defective glassy regions in crystalline In_2_O_3_ [65]. According to Preissler and Bierwagen, the existence of doubly ionized donors best describes the ionized impurity scattering in unintentionally doped In_2_O_3_ [23]. Attributing this circumstance to the existence of an indium excess, it is not unlikely to suggest such a defect as In_O_^••^, which is the non-oxidised indium at the oxygen site. It means that we might basically have In_In_^×^ − In_O_^••^ − In_In_^×^ clusters with an effective In^+2^ oxidation state.

As for oxygen vacancies, the main disagreement in literature concerns their energy level, which represents either deep [76] or shallow states [23,67]. A practical way to discover which point defects provide conductivity in oxide materials is to measure the conductivity or better the *N*_e_ dependence on *p*(O_2_). The main restriction on that is the requirement of an equilibrium, which for metal oxides means quite high temperatures, far beyond the typical 200–250 °C for In_2_O_3_:H. So the measurements performed at 800 °C discovered the *σ*
∝
*p*(O_2_)^1/10^ dependence for In_2_O_3_. Authors attributed this dependence to the (In*_i_*^•••^-O*_i_*″)^•^ cluster formation [77]. In other work Hall measurements are presented for In_2_O_3_ films obtained at different *p*(O_2_) by RF sputtering without intentional heating [78]. Conductivity was found to be rather constant (~3 × 10^3^ Ω^−1^cm^−1^) at low oxygen partial pressure (<8 × 10^−4^ Pa). When *p*(O_2_) increases, conductivity sharply drops over some orders of magnitude that is mostly caused by a decrease of *N*_e_ from ~10^−19^ to 10^−16^ cm^−3^ (at *p*(O_2_) ≈ 10^−3^ Pa). The authors suggested oxygen vacancies as the major donor defects. In this case the *N*_e_ should have revealed the slope ∝ *p*(O_2_)^−1/6^ for the very deficient oxide and ∝ *p*(O_2_)^−1/4^ for the almost stoichiometric one. From the data presented in this paper one can derive a *N*_e_
∝
*p*(O_2_)^−9^ correlation, which cannot be explained by the defect chemistry. This can be rather easier attributed to the presence of a metallic indium phase. There is another important point supporting this hypothesis. The work function, measured for metallic indium, varies in the range from ~3.9 to ~4 eV, depending on temperature [79]. This value is very close to the one measured for In_2_O_3_:H_2_O and slightly smaller as compared to the one measured for In_2_O_3_:H films (see Figure 7). This means that free electrons can easily be injected from the In^0^ outer shell into the conduction band of both oxides.

We would like to point also at the very interesting effect of photo-induced change in reactively DC-sputtered amorphous In_2_O_3_ films: an exposure of ≤100 nm thick films to UV light (*hν* ≥ 3.0 eV) resulted in a stable increase of conductivity by × 10^8^ reaching σ ≥ 10^3^ Ω^−1^cm^−1^ [80]. Simultaneously, the absorption coefficient increases by up to a factor of 10^3^ for *hv* < 1.5 eV and the absorption edge shifts by +0.1 eV. A Drude approximation of the optical absorption in the near IR region gives *N*_e_ = 1.5 × 10^20^ cm^−3^ that agrees with the Hall data. These data actually represent the pure effect of In_2_O_3_ reduction without hydrogenation/hydroxylation impact. They reproduce to some extent our results; however, the Burstein-Moss shift observed in presence of hydrogen is about 0.1 larger.

### 4.4. High-Mobility of In_2_O_3_:H

After T. Koida, the high mobility In_2_O_3_ is widely accepted to be doped by hydrogen. He also stated that the doubly ionised impurities were exchanged by singly ionised ones during the annealing process that results in about twofold reduction of *N*_e_ [44]. The in-situ Hall measurements performed by H. F. Wardenga et al. during annealing of as-deposited In_2_O_3_:H films in vacuum have allowed underlining the following stages [9]. The first stage elapsing at about 160 °C is accompanied with a slight decrease of μ_e_, which occurs, as supposed, due to the phonon scattering being expected for the degenerated semiconductors. Within this stage *N*_e_ remains settled. During further heating from 160 °C up to 200 °C, the *N*_e_ increases and μ_e_ remains unchanged. At T > 250 °C, *N*_e_ starts declining fast and a strong increase of μ_e_ takes place. The authors suggest that the driving force of the rising *N*_e_ is crystallisation followed by the grain growth. In turn, the depletion (decrease of *N*_e_) at grain boundaries is to be the reason of a measurable depletion in a material with small grains. Crystallisation and grain growth are superimposed by the decomposition of In(OH)_3_. According to the authors, the release of oxygen is responsible for the drop in carrier concentration and the grain boundaries are being saturated by hydrogen, closing dangling bonds.

We may not fully agree with this explanation mainly because of the known fact that hydrogen disappears first from the inter-grain space. We believe therefore that the dangling bonds existing at grain boundaries are most probably eliminated by the reaction (4). Moreover, it is known that the undoped single crystalline In_2_O_3_ reveals electron mobility exceeding 200 cm^2^/Vs that is restricted by ~270 cm^2^/Vs due to the phonon scattering [23]. On the other hand the unintentionally doped polycrystalline samples can demonstrate similarly high μ_e_ as the hydrogen doped ones [44,81]. According to T. Koida the effective mass in In_2_O_3_:H seems to depend on *N*_e_ mostly, rather than on crystallinity [44]. Many of the experimental data collected for various In_2_O_3_ based systems reveal a plateau on μ_e_ = *f*(*N*_e_) dependency exactly around *N*_e_ ~ 10^20^ cm^−3^ [23]. This phenomenon is also associated with a large spread of mobilities indicating additional scattering due to imperfections in the crystal for the samples with μ_e_ < 130 cm^2^/Vs.

## 5. Conclusions

To conclude, we observed that the free charge carriers in both In_2_O_3_ and In_2_O_3_:H films can appear due to the presence of In^0^. We suggest that metallic indium is present in as-deposited In_2_O_3_ or In_2_O_3_:H_2_O films in a much, up to the atomic level, dispersed state. The presence of water or hydrogen during In_2_O_3_ deposition at low temperature secures the amorphous state of the film. Hydroxylation of In_2_O_3_ is probably the main reason for that. Crystallisation of such films starts at ~160 °C when the excess of indium agglomerates, releasing in a separate nano-crystalline phase due to the melting. Melted indium species may vanish during annealing in two ways: either via evaporation and oxidation by water in UHV or via oxidation by oxygen in air. Thus, the concentration of free electrons in In_2_O_3_ matrix is reduced and the near IR transparency increases. Both processes, however, do not provide high mobility. The laterally extended growth of crystallites happens when water is released as a result of the hydroxide → oxide transformation. Growing crystallites interconnect at grain boundaries by the In-O-In bonds. These factors both provide high electron mobility exceeding 100cm^2^/Vs. According to our experimental observation, annealing in air demands lower temperature (~180 °C) to provide high mobility as compared to the annealing in UHV (>220 °C). We attribute this to the higher water content in the former case. Crystallisation of the In_2_O_3_:H_2_O system is accompanied with the doping of In_2_O_3_. We expect that the “unintentional” doping differs from the “hydrogen” doping as follows. In the first case a spontaneous injection of free charge carriers from the dispersed In^0^ metallic species concentrated in the inter-grain defect-rich spaces takes place. In the second case, we likely deal with oxidation of the intra-grain In^0^ defects trapped during crystallisation by Schottky vacancies:In_O_^••^ + ½ O_2_ + V_In_‴ → In_In_^×^ + O_O_^×^ + *e*′(5)
In_O_^••^ + H_2_O + V_In_‴ → In_In_^×^ + OH_O_^•^ + ½ H_2_ + 2*e*′(6)

Reactions (5) and (6) describe oxidation by oxygen and water, respectively. High temperature makes water an oxidizing agent, whereas low pressure facilitates hydrogen removal. According to our SIMS results, gaseous hydrogen is removed from the film mostly from the top ~50 nm layer. Metallic indium can also accumulate hydrogen in the bulk of the film. As we saw, OH_O_^•^ defects most probably also exist in In_2_O_3_:H films but their formation in the absence of In^0^ is not associated with any redox reaction, so in that case they do not donate electrons.

## Figures and Tables

**Figure 1 materials-12-00266-f001:**
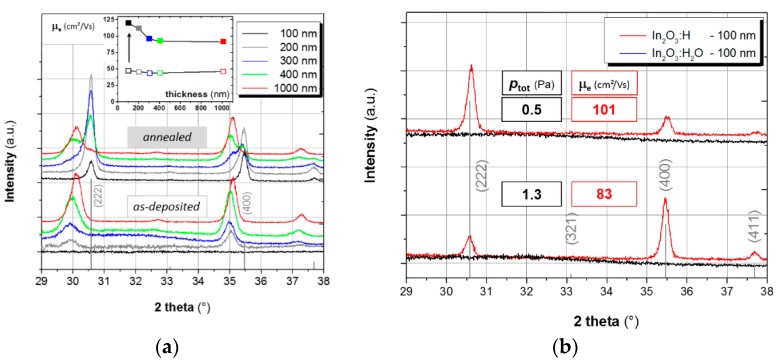
XRD patterns and Hall-mobility data for the RF sputtered In_2_O_3_:H films: (**a**) variation of thickness at *p*_tot_ = 0.5 Pa; (**b**) variation of the sputtering pressure at a fixed film thickness. In all cases the as-deposited state (In_2_O_3_:H_2_O) is compared with the annealed one (In_2_O_3_:H). The cubic In_2_O_3_ powder standard (ICDD Nr. 00-006-0416) is shown in grey bars.

**Figure 2 materials-12-00266-f002:**
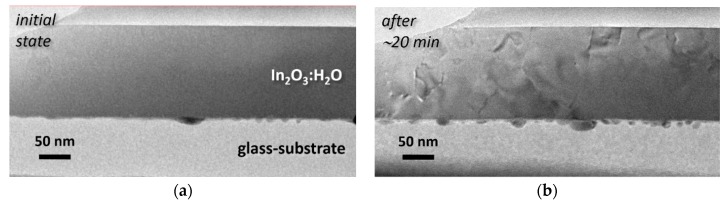
Brightfield TEM images of the as-deposited In_2_O_3_:H_2_O film: (**a**) first minutes of observation; (**b**) after about 20 min under the electron beam.

**Figure 3 materials-12-00266-f003:**
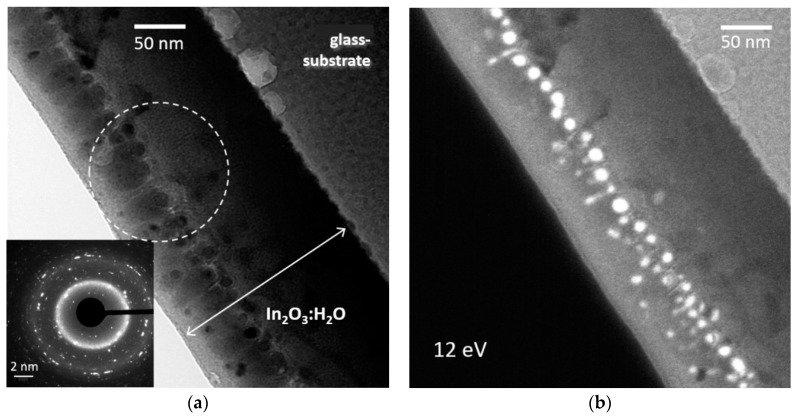
Medium magnification bright field image (**a**) and EFTEM image at an electron energy loss of 12 eV (**b**) obtained on an as-deposited In_2_O_3_:H_2_O film at the same location. Image (**a**) elucidates two parts of the film: the bottom ~100 nm thick amorphous part and the top ~50 nm thick crystalline part. The electron diffraction pattern is obtained for the region, marked by a circle. Bright regions in the figure (**b**) correspond to metallic indium.

**Figure 4 materials-12-00266-f004:**
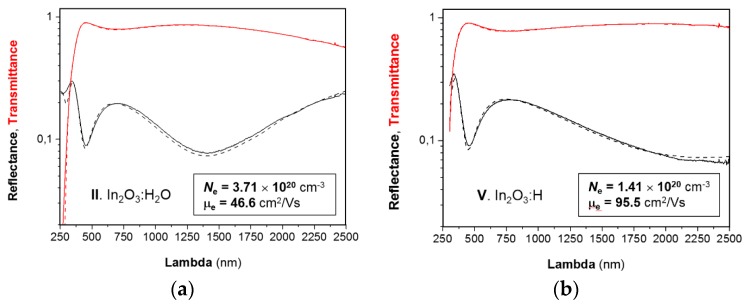
Fits (dashed lines) of the optical spectra (solid lines) for ~100 nm thick In_2_O_3_:H_2_O (**a**) and In_2_O_3_:H (**b**) films on glass. The μ_e_ and *N*_e_ values identified from the fitted spectra are given. Corresponding plasma edges, film thicknesses and mean square errors are as follows: (**a**) λ_p_ = 916.918 nm, *d* = 103.385 nm, MSE = 0.00757124; (**b**) λ_p_ = 1486.62 nm, *d* = 104.554 nm, MSE = 0.00893229.

**Figure 5 materials-12-00266-f005:**
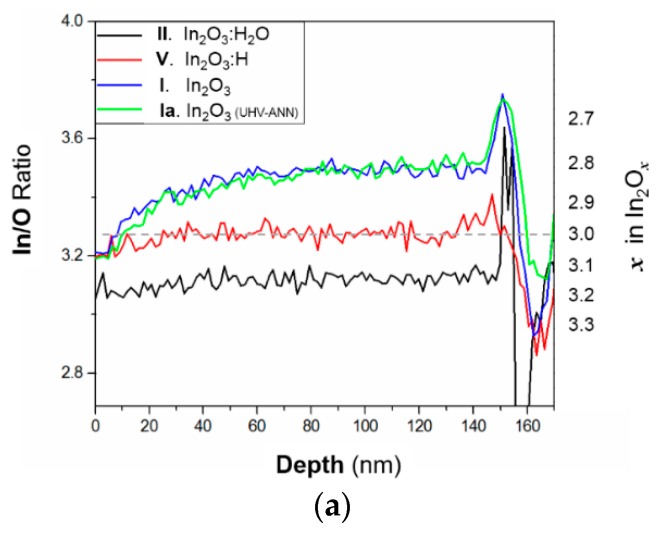
SIMS depth profiles obtained on various ~150 nm thick DC sputtered In_2_O_3_ films. Indium to oxygen concentration ratios (**a**) were obtained for the film states I, II and V, which correspond to the Table 1. State Ia represents the UHV-annealed In_2_O_3_ film. Hydrogen to oxygen concentration ratio (**b**) and the percentage losses of hydrogen and oxygen (**c**) are compared for the as-deposited and annealed states of the film, intentionally containing water.

**Figure 6 materials-12-00266-f006:**
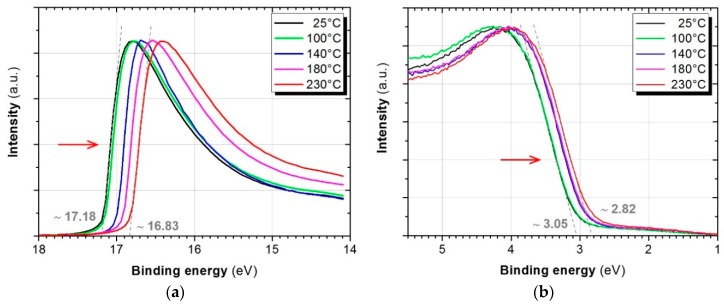
UPS spectra obtained for the ~150 nm thick In_2_O_3_:H_2_O films during stepwise annealing in UHV without breaking vacuum. Two regions (**a**,**b**) of the same spectra are shown. For comparison, the intensity was normalised and the background was removed. The excitation energy was 21.2 eV.

**Figure 7 materials-12-00266-f007:**
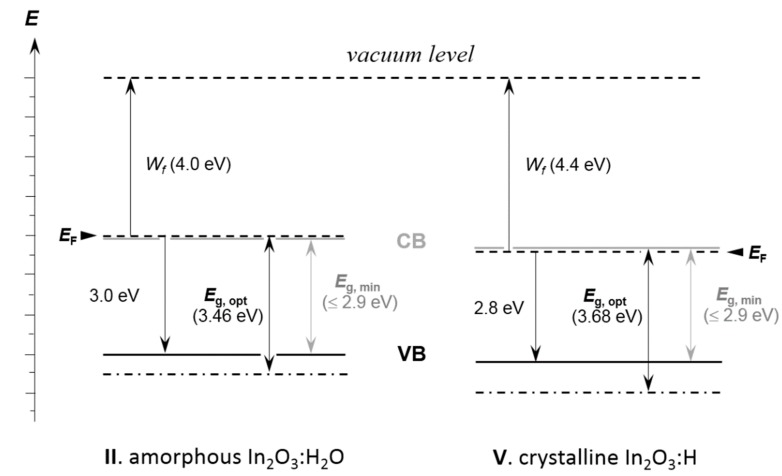
Energy diagram for indium oxide of the states II and V. The scheme is created on the basis of optical (UV-Vis) and UPS data.

**Table 1 materials-12-00266-t001:** Results of the XRD analysis and electron Hall-mobility data for the differently prepared ~150 nm thick In_2_O_3_ films on glass. The Roman numerals are used in the text for easier distinction of the film states.

States	I. In_2_O_3_ As-Deposited w/o Heating	II. In_2_O_3_:H_2_O As-Deposited w/o Heating	III. In_2_O_3_:H_2_O As-Deposited T*_dep_* = 160 °C	IV. In_2_O_3_:H T*_ann_* = 180 °C, Ambient Air	V. In_2_O_3_:H T*_ann_* = 230 °C, UHV
*D_long_* (nm)	35 ± 6	-	50 ± 15	165 ± 6	328 ± 5 *
*D_lat_* (nm)	22 ± 2	-	32 ± 2	230 ± 60	205 ± 20
*Residual Stress* (GPa)	−2.1 ± 0.8	-	−1.9 ± 0.6	−0.5 ± 0.1	−0.6 ± 0.2
μ_e_ (cm^2^/V⋅s)	22	47	41	117	118
*N*_e_ (cm^−3^)	4.58 × 10^20^	5.85 × 10^20^	2.03 × 10^20^	2.24 × 10^20^	2.57 × 10^20^

* This value has no real physical meaning, because the film is considerably thinner.

**Table 2 materials-12-00266-t002:** Results obtained from the optical absorption of variously prepared In_2_O_3_ films.

States → Band Gap/Thickness ↓	I. In_2_O_3_ As-Deposited w/o Heating	II. In_2_O_3_:H_2_O As-Deposited w/o Heating	V. In_2_O_3_:H T*_ann_* = 230 °C, UHV
*E*_g_ (eV)/100 nm 400 nm	3.55 3.50	3.46 3.48	3.68 3.65

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
