# Peer review of "Crystallisation Phenomena of In2O3:H Films"

_materials, 2019, doi:10.3390/ma12020266_

Reviewer 1 Report

Major translation revision are required.

The written English needs extensive editing, to the point that I cannot read the manuscript clearly.  I am not sure what the authors are trying to say and explain in most situations.  The explanations of the diagrams are incoherent (Figure 1) and it seems there are paragraphs (lines 167-170 for example) which are completely out of context and do not seem to follow from the previous discussion.  

For this reason, I am unable to adequately evaluate this manuscript in its present form, and I find it not ready for publication.  I suspect there might be some useful and valid information here, but I cannot tell with the extensive translation edits necessary for comprehension. 

I sympathize with the authors trying to write in a different language, but sentence structure and format are not correct.

Author Response

Thank you. We must admit that you are right. This manuscript was extended stepwise over two years. Finally the starting sections were written unsatisfactory. We have done our the best to improve English and clarity.

Reviewer 2 Report

This manuscript reportd the crystallization phenomena of In2O3:H films.

the provided data, captions and explains are not clear and enough to support the conclusions. In this form, this manuscript can not be suitable for the publication.

1). You described the films of In2O3:H2O having H2O. Is H2O stable in Plasma? Can H2O be incorporated into In2O3 film as H2O itself?

2). In Table 1, Table 2,  and Fig 2, the thickness of films looks like 150nm. However, in Fig. 1, there were no xrd data about 150nm thick films. Pleases, provide the xrd of it.

3). Please, provide the exact status of films in Fig 2. and Fig 3 such as growth condition, thickness, and post-heat treatment.  the thickness of the films between Fig 2 and Fig 3. were different. In Fig 3. the film looks uniform amorphous in fig 2a and changed to the crystallization happened in all ares though the thickness in fig2b. But, in Fig 3. the film has the top crystalline part and the bottom amorphous part. What is the difference in films? w

4).  In caption of Table2, the authors mention the 150nm thickness of films. But, inside of table, the authors showed 100nm, and 200nm thickness. Make it clear.

5) In Fig5. the authors showed the data from the films deposited by DC sputtering. However, the data of Fig. 1, 2, 3, and4 were obtained from the films deposited by RF sputtering. For the consistency in discussion, the same growth methods are recommended.

6) In abstract, the author mentioned “This was considered as a likely intra-grain phenomenon caused by both (In0)O and (OH-)Opoint defects” in line 29 and 30. Howver, there were no discussion about the (Ino)O.., non-oxidized indium at the oxygen site.  Is it possible? The author mentioned the possibility of the cluster in line 493. You need the more evidence and discussion for it.

6) In conclusion,

In equation 4 and 5, non-oxidized indium at the oxygen site was included. the authors must choose one of them. If the authors want to claim the defect of the non-oxidized indium at the oxygen site, the more evidence and discussion are needed.

Author Response

ANSWER 1:

The definition: “high mobility hydrogen doped In2O3:H” is widely accepted in literature and stays for the annealed state of In2O3 film being sputtered in presence of water vapors. The as-deposited amorphous state reveals OH-groups in XPS spectra [Ref.9, our data]. This signal disappears after annealing. Basically, metal oxides form hydroxides binding water:

In2O3 + 3 H2O -> 2 In(OH)3

Exactly this basic fact is expressed in our formal definition “”In2O3:H2O” just to point at presence of hydroxides. Water content is variable and depends on deposition conditions. Plasma chemistry of water is discussed in the subsection Appearance of metallic indium in In2O3 . In fact, water partially dissociates and ionises in Ar-plasma, but then, in a solid In2O3 film at room temperature it can present only in a stable form, like indium hydroxide or bare H2O.

If Reviewer would recommend another term for the state in question, please propose. We find the following ones also acceptable: In2O3:In(OH)3; InxOy(OH)z or use quotes for the “In2O3:H2O”.

ANSWER 2:

We have extended the supplementary materials with these data.

ANSWER 3:

Here we deal with RF sputtered (ptot = 0.5 Pa) In2O3:H2O (not annealed) film. The sample presented in the figures 2, 3, S3, S4 and S5 is the same, however we investigated two lamella. First lamella was unexpectedly partially crystallized during TEM investigation (Fig. 2). Then we provided better heat dissipation via specimen holder in another TEM system with EELS. Figures 3, S3, S4 (b) and S5 were obtained from the different places of the lamella No.2. Fig. S2 (a) shows the remaining amorphous part of the lamella No.1.

This film was expected to be 150 nm thick based on our profiler data. Evidently, the thickness is not constant over the film, additionally, crystallisation in the top layer partly observed. We suggest these peculiarities to relate to the effect of magnetron. Namely, the regions of the film which were directly opposite to the racetracks of electrons should be heated by plasma more impactful -> thicker, partially crystalline film. An example of such inhomogeneity is presented in our previous paper [Ref.13].

We added these circumstances to the text in supplementary materials.

ANSWER 4: Thank you.

ANSWER 5: Thank you. We have experienced for several years sputtering of “high mobility hydrogen doped In2O3” by three different systems (x2 RF and DC). We observed the identity of optical and electrical properties between DC- and RF-films if the as-deposited films were obtained in an amorphous state. The only critical impact resulting in irreproducibility is a partial crystallization taking place during deposition if the substrate is overheated by plasma.

ANSWER 6: We just suggest the presence of (In0)O·· defects being guided by the following considerations: (i) proved existence of metallic indium in the films in question; (ii) contradiction of various DFT calculations in literature; (iii) conductivity dependence on p(O2) is not widely reproduced and the majority of experimental data cannot be explained by the formation of classic defects like VO··; (iv) much better size fitting, because In0 is rather large to sit at the In3+-site.

ANSWER 7: The reason to present these (4) and (4a) reactions in such form is to demonstrate that independently on In0 position, only H2O can oxidize it with hydrogen release.

GENERAL: In spite of quite critical response, we thank the reviewer for being specific. This allowed us to improve our current texts and to learn for the future.

Reviewer 3 Report

The presented work give valuable insights for the understanding of material changes during In2O3:H crystallization. The presence and influence of In in both In2O3 and In2O3:H films were properly measured and discussed in details. Despite off all benefits some small improvements could enhance added value of presented work:

The presented work enhance our understanding on the material changes during In2O3 crystallization mostly on theoretical level. Would be useful to add few thoughts how this research could positively impact applicability of In2O3:H thin films in Introduction part.   

SEM measurements were mentioned in lines 116-117 and 578-579. Despite of it, SEM measurements were not included in manuscript. Would be useful to exclude this information from manuscript.

Author Response

Thank you. We have inserted the requested SEM picture in supplementary materials.

Reviewer 4 Report

According to manuscript Materials-402600, entitled „Crystallisation Phenomena of In2O3:H Films“ by authors R. Muydinov, A. Steigert, M. Wollgarten, P. P. Michałowski, U. Bloeck, A. Pflug, D. Erfurt, R. Klenk, S. Körner, I. Lauermann, B. Szyszka

The manuscript deals with the investigation of sputtered, amorphous In2O3:H films and their crystallization process. The sputtered films have been studied depending on the film thickness, the process pressure and the water presence during deposition. Post-deposition low temperature annealing is also applied. Good work, systematic investigation and very in depth discussions. The structural, electrical and optical properties are considered.

The paper is suitable for publication and can be accepted in its present form.

Author Response

Thank you.

Reviewer 5 Report

Manuscript ID: materials-402600

Title: Crystallisation Phenomena of In2O3:H Films

Comments:

The authors describe the crystallization behavior of sputtered In2O3:H films by thermal annealing. Various characterization methods were used to figure out the crystallization process of the In2O3:H films. The In2O3:H films as a window layer are very promising in Si-based and CIGS solar cells, due to their higher mobility, improved NIR transmittance, and higher work function compared to other TCO films. So, such a basic research work can have an impact for material engineers to better understand the material physics and chemistry for device application. However, the reviewer has some specific comments on their manuscript to improve the contents.

. The last paragraph of introduction needs to add what you did and what you resulted shortly.

. The reviewer suggests having a specific table for materials deposited in the experimental.

. The manuscript seems to be generally too long so that please, try to shorten all sections.

. In addition, the reviewer finds a lot of sentences formulated a bit cloudy.

Author Response

ANSWERS: Thank you. We enhanced the introduction. Such a table with all samples-methods-analyses would be a bit confusing to our opinion, because the reader will be guided to jump often over the manuscript. We hopefully improved the description of samples. We have tried to improve the logics of the text, however, it was hard to make it shorter because of the following reason. The absolute majority of publications devoted to the “high-mobility hydrogen doped indium oxide” is written by physicists and many explanations they give are questionable in terms of chemistry. However, the understanding of this material claims high expertise in chemistry as well. That is why we tried to review all aspects which may be relevant. Our manuscript is a kind of puzzle which we compose from the numerous published data and our experimental findings.

Round  2

Reviewer 5 Report

Manuscript ID: materials-402600

Title: Crystallisation Phenomena of In2O3:H Films

Comments:

The authors provide a proper response to the comments. The manuscript is good now for the publication in Materials Journal as is.